# Unsupervised Generative 3D Shape Learning from Natural Images

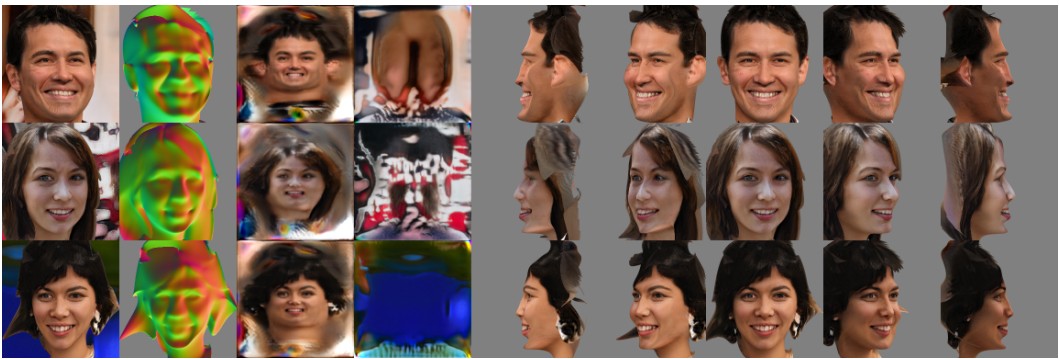

Figure 1: Samples from our generator trained on the FFHQ dataset at $128 \times 128$ resolution. The first column shows random rendered samples. The other columns show the 3D normal map, texture, background and textured 3D shapes for 5 canonical viewpoints in the range of $\pm 90$ degrees.

## Abstract

In this paper we present, to the best of our knowledge, the first method to learn a generative model of 3D shapes from natural images in a fully unsupervised way. For example, we do not use any ground truth 3D or 2D annotations, stereo video, and ego-motion during the training. Our approach follows the general strategy of Generative Adversarial Networks, where an image generator network learns to create image samples that are realistic enough to fool a discriminator network into believing that they are natural images. In contrast, in our approach the image generation is split into 2 stages. In the first stage a generator network outputs 3D objects. In the second, a differentiable renderer produces an image of the 3D objects from random viewpoints. The key observation is that a realistic 3D object should yield a realistic rendering from any plausible viewpoint. Thus, by randomizing the choice of the viewpoint our proposed training forces the generator network to learn an interpretable 3D representation disentangled from the viewpoint. In this work, a 3D representation consists of a triangle mesh and a texture map that is used to color the triangle surface by using the UV-mapping technique. We provide analysis of our learning approach, expose its ambiguities and show how to overcome them. Experimentally, we demonstrate that our method can learn realistic 3D shapes of faces by using only the natural images of the FFHQ dataset.

## 1 Introduction

Generative Adversarial Nets (GAN) (see Goodfellow et al. (2014)) have become the gold standard generative model over the years. Their capability is demonstrated in many data sets of natural images. These generative models can create sample images that are nearly indistinguishable from real ones. GANs do not need to make assumptions on the data formation other than applying a neural network on a latent noise input. They can also operate in a fully unsupervised way. GANs have strong theoretical foundations since the beginning. Goodfellow et al. (2014) demonstrate that,

under suitable conditions, GANs learn to generate samples with the same probability distribution as samples in the training data set have.

However, there is one notable drawback of GANs compared to classical generative models like Gaussian Mixture Models by Xu & Jordan (1996) or Naive Bayes classifiers by McCallum et al. (1998). Classical models are interpretable, but GANs are not. Interpretability is achieved in classical models by making strong assumptions on the data formation process and by using them on interpretable engineered features.

Our work combines the best of both worlds for 3D shape learning. We keep the advantages of GANs: unsupervised training, applicability on real datasets without feature engineering, the theoretical guarantees and simplicity. We also make our representation interpretable, as our generator network provides a 3D mesh as an output. To make the learning possible, we make the assumption that natural images are formed by a differentiable renderer. This renderer produces fake images by taking the 3D mesh, its texture, a background image and a viewpoint as its input. During training a discriminator network is trained against the generator in an adversarial fashion. It tries to classify whether its input is fake or comes from the training dataset.

The key observation is that a valid 3D object should look realistic from multiple viewpoints. This image realism is thus enforced by the GAN training. This idea was first applied for generating 3D shapes in Gadelha et al. (2017). Their pioneering work had several limitations though. It was only applicable to synthetic images, where the background mask is available and produced only black and white images. Our method works on natural images and we do not need silhouettes as supervision signal.

From a theoretical point of view, image realism means that the images lie inside the support of the probability distribution of natural images (see Ledig et al. (2017), where image realism was used for super-resolution). AmbientGAN of Bora et al. (2018) proves that one can recover an underlying latent data distribution, when a known image formation function is applied to the data, thus achieving data realism by enforcing image realism. Our work and that of Gadelha et al. (2017) are special cases of AmbientGAN for 3D shape learning. However this task in general does not satisfy all assumptions in Bora et al. (2018), which results in ambiguities in the training. We resolve these issues in our paper by using suitable priors.

We summarize our contributions below:

- For the first time, to the best of our knowledge, we provide a procedure to build a generative model that learns explicit 3D representations in an unsupervised way from natural images. We achieve that using a generator network and a renderer trained against a discriminator in a GAN setting. Samples from our model are shown in Figure 1.

- We introduce a novel differentiable renderer, which is a fundamental component to obtain a high-quality generative model. Notably, it is differentiable with respect to the 3D vertex coordinates. The gradients are not approximated, they can be computed exactly even at the object boundaries and in the presence of self-occlusions.

- We analyze our learning setup in terms of the ambiguities in the learning task. These ambiguities might derail the training to undesirable results. We show that these problems can only be solved when one uses labels or prior knowledge on the data distribution. Finally, we provide practical solutions to overcome the problems that originate from the ambiguities.

## 2  RELATED WORK

Before we discuss prior work, we would like to state clearly what we mean by supervised, unsupervised and weakly supervised learning in this paper. Usually, supervised learning is understood as using annotated data for training, where the annotation was provided by human experts. In some scenarios the annotation comes from the image acquisition setup. Often these approaches are considered unsupervised, because the annotation is not produced by humans. Throughout our paper we consider supervision based on the objective function and its optimization, and not based on the origin of the annotation. To that purpose, we use the notion of the target and training objective. The *target objective* is defined as the function that measures the performance of the trained model $f$, *i.e.*,

$$\mathcal{L}_{\text{target}}(f) = \mathbb{E}[l_t(f(\mathbf{x}), \mathbf{y})], \tag{1}$$

Table 1: Comparison of prior work. We indicated all supervision signals for the full training process. In case of methods that work on synthetic images with static background we indicated silhouettes as supervision signal.

| 3D Method | Supervision Signal | | | | | | 3D Representation | | | | Capabilities | |
|---|---|---|---|---|---|---|---|---|---|---|---|---|
| | Images | 3DMM | 3D | Keypoints | Silhouettes | Viewpoint | Point Cloud | Voxel | Mesh | Texture | View Control | Natural Images |
| Geng et al. (2019) | ✓ | ✓ | ✓ | ✓ | | | | | ✓ | ✓ | | ✓ |
| Gecer et al. (2019) | ✓ | ✓ | | ✓ | | | | | ✓ | ✓ | ✓ | ✓ |
| Sanyal et al. (2019) | | ✓ | | ✓ | | | | | ✓ | ✓ | ✓ | ✓ |
| Ranjan et al. (2018) | | | ✓ | | | | | | ✓ | | | |
| Kato & Harada (2019) | ✓ | | | | ✓ | ✓ | | | ✓ | ✓ | | |
| Achlioptas et al. (2018) | | | ✓ | | | | ✓ | | | | | |
| Wu et al. (2016) | | | ✓ | | | | | ✓ | | | | ✓ |
| Paysan et al. (2009) | | | ✓ | | | | | | ✓ | ✓ | ✓ | ✓ |
| Gerig et al. (2018) | | | ✓ | | | | | | ✓ | ✓ | ✓ | ✓ |
| Gadelha et al. (2017) | | | | | ✓ | | ✓ | | | | ✓ | |
| Rajeswar et al. (2019) | ✓ | | | | | | | | ✓ | | ✓ | |
| Rezende et al. (2016) | ✓ | | | | | | | ✓ | ✓ | ✓ | ✓ | |
| Henzler et al. (2018) | ✓ | | | | ✓ | | | ✓ | | ✓ | ✓ | ✓ |
| Henderson & Ferrari (2019) | ✓ | | | | ✓ | | | | ✓ | | ✓ | |
| Nguyen-Phuoc et al. (2019) | ✓ | | | | | | | | | | ✓ | ✓ |
| **This work** | ✓ | | | | | | | | ✓ | ✓ | ✓ | ✓ |

where $l_t$ is the loss function, $\mathbf{x}$, $\mathbf{y}$ are the data and labels respectively. The *training objective* is the function that is optimized during the training, They are defined for the supervised, weakly supervised and unsupervised case as

$$\mathcal{L}_{\text{supervised}}(f) = \mathbb{E}[l_s(f(\mathbf{x}), \mathbf{y})], \tag{2}$$

$$\mathcal{L}_{\text{weakly}}(f) = \mathbb{E}[l_w(f(\mathbf{x}), \mathbf{y}_w)], \tag{3}$$

$$\mathcal{L}_{\text{unsupervised}}(f) = \mathbb{E}[l_u(f(\mathbf{x}))], \tag{4}$$

where $\mathbf{y}_w$ denotes a subset of labels and the loss functions $l_s$, $l_w$ and $l_u$ may be different from $l_t$. In the case of most supervised tasks (*e.g.* in classification) the target is the same as the training objective (cross-entropy). Another example is monocular depth estimation. In this case, the inputs are monocular images and the model can be trained using stereo images. This makes it a weakly supervised method under the definitions above. The GAN training is unsupervised, it has the same target and training objectives (the Jensen-Shannon divergence), and thus even the target objective does not use labels. In Table 1, we show the most relevant prior work with a detailed list of used supervision signals. There, we consider the full training scenario from beginning to end. For example if a method uses a pre-trained network from another previous work in its setup, we consider it supervised if the pre-trained network used additional annotation during its training.

A very successful 3D generative model is the Basel face model introduced by Paysan et al. (2009), which models the 3D shape of faces as a linear combination of base shapes. To create it, classical 3D reconstruction techniques (see Hartley & Zisserman (2003)) and laser scans were used. This model is used in several methods (*e.g.* Geng et al. (2019); Gecer et al. (2019); Sanyal et al. (2019); Sela et al. (2017); Tran et al. (2017); Genova et al. (2018); Tewari et al. (2017)) for 3D reconstruction tasks by regressing its parameters. There are other methods based on GANs, such as those of Wu et al. (2016); Achlioptas et al. (2018), and autoencoders (*e.g.* Ranjan et al. (2018)) that learn 3D representations by directly using 3D as the supervision signal.

The most relevant papers similar to our work are the ones that use differentiable rendering and using randomly sampled viewpoints to enforce image realism. There are GAN based methods by Gadelha et al. (2017); Henzler et al. (2018); Rajeswar et al. (2019) and Variational autoencoder based methods by Rezende et al. (2016); Henderson & Ferrari (2019); Kato & Harada (2019). However,

Figure 2: Illustration of the training setup. $G$ and $D$ are the generator and discriminator neural networks. $R$ is the differentiable renderer and it has no trainable parameters. The random variables $\mathbf{z}$, $\mathbf{m}$ and $\mathbf{v}$ are the latent vector, 3D object and the viewpoint parameters. The fake images are $\mathbf{x}_f$ and the real images are $\mathbf{x}_r$.

these methods are only applicable on synthetic data or use weak supervision for training as shown in Table 1.

Our method can also be interpreted as a way to disentangle the 3D and the viewpoint factors from images in an unsupervised manner. Reed et al. (2015) used image triplets for the task. They utilized an autoencoder to reconstruct an image from the mixed latent encodings of other two images. Mathieu et al. (2016) and Szabó et al. (2018) only use image pairs that share the viewpoint attribute, thus reducing part of the supervision in the GAN training. Similarly, StyleGAN Karras et al. (2019) and Hu et al. (2018) use mixing latent variables for unsupervised disentangling. HoloGAN (Nguyen-Phuoc et al. (2019)) is a method for disentangling objects and viewpoints using latent shape representations. In contrast we learn an explicit 3D mesh representation, which can be used in traditional rendering pipelines. By using this rendering in the training, we demonstrate the disentangling of the 3D shape from the viewpoint without any labels and also guarantee interpretability and consistency across viewpoints.

An important component of our model is the renderer. Differentiable renderers like Neural mesh renderer Kato et al. (2018) or OpenDR Loper & Black (2014). have been used along with neural networks for shape learning tasks. Differentiability is essential to make use of the gradient descent algorithms commonly employed in the training of neural networks. We introduce a novel renderer, where the gradients are exactly computed and not approximated at the object boundaries.

## 3 METHOD

We are interested in building a mapping from a random vector to a 3D object (texture and vertex coordinates), and a background image. We call these three components the *scene representation*. To generate a view of this scene we also need to specify a viewpoint, which we call the *camera representation*. The combination of the scene and camera representations is also referred to as simply the *representation*, and it is used by a differential renderer $R$ to construct an image.

We train a generator $G$ in an adversarial fashion against a discriminator $D$ by feeding zero-mean Gaussian samples $\mathbf{z}$ as input (see Fig. 2). The objective of the generator is to map Gaussian samples to scene representations $\mathbf{m}$ that result in realistic renderings $\mathbf{x}_f$ for the viewpoint $\mathbf{v}$ used during training. The discriminator then receives the fake $\mathbf{x}_f$ and real $\mathbf{x}_r$ images as inputs. The GAN training solves the following optimization problem,

$$\min_{G} \max_{D} \ \mathbb{E}_{\mathbf{x}_r \sim p_r}[\log(D(\mathbf{x}_r))] + \mathbb{E}_{\mathbf{z} \sim \mathcal{N}, \mathbf{v} \sim p_{\mathbf{v}}}[\log(1 - D(R(G(\mathbf{z}), \mathbf{v})))], \tag{5}$$

where $\mathbf{x}_f = R(G(\mathbf{z}), \mathbf{v})$ are the generated fake images, $\mathbf{m} = G(\mathbf{z})$ are the 3D shape representations and $\mathbf{x}_r$ are the real data samples. The renderer $R$ is a fixed function, *i.e.*, without trainable parameters, but differentiable. The viewpoints $\mathbf{v}$ are randomly sampled from a known viewpoint distribution. In practice, $G$ and $D$ are neural networks and the optimization is done using a variant of stochastic gradient descent (SGD).

## 4 THEORY

In this section we give a theoretical analysis of our method and describe assumptions that are needed to make the training of the generator succeed. We build on the theory of Bora et al. (2018) and examine its assumptions in the 3D shape learning task.

**Assumption 1** *The images in the dataset* $\mathbf{x}_r = R(\mathbf{m}_r, \mathbf{v}_r)$ *are formed by the differentiable rendering function* $R$ *given the 3D representation* $\mathbf{m}_r \sim p_{\mathbf{m}}$ *and viewpoint* $\mathbf{v}_r \sim p_{\mathbf{v}}$.

Here $p_{\mathbf{m}}$ and $p_{\mathbf{v}}$ are the "true" probability density functions of 3D scenes and viewpoints. This assumption is needed to make sure that an optimal generator exists. If some real images cannot be generated by the renderer then the generator can never learn the corresponding model. We can safely assume we have a powerful enough renderer for the task. Note that this does not mean the real data has to be synthetically rendered with the specific renderer $R$.

**Assumption 2** *The ground truth viewpoint* $\mathbf{v}_r \sim p_{\mathbf{v}}$ *and the 3D scenes* $\mathbf{m}_r \sim p_{\mathbf{m}}$ *are independent random variables. The distribution of the 3D representations* $p_{\mathbf{m}}$ *is not known (it will be learned). The viewpoint distribution* $p_{\mathbf{v}}$ *is known, and we can sample from it, but the viewpoint* $\mathbf{v}_r$ *is not known for any specific data sample* $\mathbf{x}_r$.

This assumption is satisfied unless images in the dataset are subject to some capture bias. For example, this would be the case for celebrities that have a "preferred" side of their face when posing for pictures. More technically, this assumption allows us to randomly sample $\mathbf{v}$ viewpoints independently from the generated $\mathbf{m}$ models.

**Assumption 3** *Given the image formation model* $\mathbf{x} = R(\mathbf{m}, \mathbf{v})$, *when* $\mathbf{v} \sim p_{\mathbf{v}}$, *there is a unique distribution* $p_{\mathbf{m}}$ *that induces the data* $\mathbf{x} \sim p_{\mathbf{x}}$.

This assumption is necessary for learning the ground truth $p_{\mathbf{m}}$. In the unsupervised learning task we can only measure the success of the learning algorithm on the output images during the training. If multiple distributions can induce the same data $\mathbf{x} \sim p_{\mathbf{x}}$, there is no way for any learning algorithm to choose between them, unless prior knowledge on $p_{\mathbf{m}}$ (in practice it is an added constraint on $\mathbf{m}$ in the optimization) is available.

The 3D shape learning task in general is ambiguous. For example in the case of the hollow-mask illusion (see Gregory (1970)) people are fooled by an inverted depth image. Many of these ambiguities depend on the parametrization of the 3D representation. For example different meshes can reproduce the same depth image with different triangle configurations, thus a triangle mesh is more ambiguous than a depth map. However, if the aim is to reconstruct the depth of an object, the ambiguities arising from the mesh representation do not cause an ambiguity in the depth. We call these *acceptable ambiguities*.

For natural images one has to model the whole 3D scene, otherwise there is a trivial ambiguity when a static background is used. The generator can simply move the object out of the camera field of view and generate the images on the background like in the GAN training to generate natural images. Modelling the whole scene with a triangle mesh however is problematic in practice because of the large size of the (multiple) meshes that would need. We propose a compromise, where we only model the object with the mesh and we generate a large background and crop a portion of it randomly during training. In this way, even if the generator outputs background images with a view of the object, there is no guarantee that the background crop will still contain the object. Thus, the generator will not be able to match the statistics of real data unless it produces a realistic 3D object.

**Assumption 4** *The generator* $G$ *and discriminator* $D$ *have large enough capacity, and the training reaches the global optimum of the GAN training 5.*

In practice, neural networks have finite capacity and we train them with a local iterative minimization solver (stochastic gradient descent). Hence, the global optimum might not be achieved. Nonetheless we show that the procedure works in practice.

Now we show that under these conditions the generator can learn the 3D geometry of the scene faithfully.

**Theorem 1** *When the above assumptions are satisfied, the generated scene representation distribution is identical to the real one, thus* $G(\mathbf{z}) \sim p_{\mathbf{m}}$, *with* $\mathbf{z} \sim \mathcal{N}(0, I)$.

The proof can be readily adapted from Bora et al. (2018).

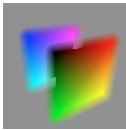

Figure 3: Illustration of our renderer. We show a rectangle that occludes another.

## 5 DIFFERENTIABLE RENDERER

Differentiability is essential for functions used in neural network training. Unfortunately traditional polygon renderers are only differentiable with respect to the texture values but not for the 3D vertex coordinates. When the mesh is shifted by a small amount, the rendered pixel can shift to the background or to another triangle in the mesh. This can cause problems during training. Thus, we propose a novel renderer that is differentiable with respect to the 3D vertex coordinates.

We make the renderer differentiable by extending the triangles at their boundaries with a fixed amount in pixel space. Then, we blend the extension against the background or against an occluded triangle. The rendering is done in two stages. First, we render an image with a traditional renderer, which we call the *crisp* image. Second, we render the triangle extensions and its alpha map, which we call the *soft* image. Finally we blend the crisp and the soft images.

We render the crisp image using the barycentric coordinates. Let us define the distance of a 2D pixel coordinate $\boldsymbol{p}$ to a triangle $T$ as $d(\boldsymbol{p}, T) = \min_{\boldsymbol{p}_t \in T} d(\boldsymbol{p}, \boldsymbol{p}_t)$ and its closest point in the triangle as $\boldsymbol{p}^*(T) = \arg\min_{\boldsymbol{p}_t \in T} d(\boldsymbol{p}, \boldsymbol{p}_t)$, where $d$ is the Euclidean distance. For each pixel and triangle the rendered depth, attribute and alpha map can be computed as

$$z_c(\boldsymbol{p}, T) = \mathbf{1}\{d(\boldsymbol{p}, T) = 0\} \sum_{i=1}^{3} b_i(\boldsymbol{p}, T) z_i(T) + (1 - \mathbf{1}\{d(\boldsymbol{p}, T) = 0\}) z_{far} \tag{6}$$

$$a_c(\boldsymbol{p}) = \mathbf{1}\{z_c(\boldsymbol{p}, T_c^*(\boldsymbol{p})) < z_{far}\} \sum_{i=1}^{3} b_i(\boldsymbol{p}, T_c^*(\boldsymbol{p})) a_i(T_c^*(\boldsymbol{p})) \tag{7}$$

$$\alpha_c(\boldsymbol{p}) = \mathbf{1}\{z_c(\boldsymbol{p}, T_c^*(\boldsymbol{p})) < z_{far}\} \tag{8}$$

where $z_c$ indicates the depth of the crisp layer, $b_i$ are the barycentric coordinates and $z_i(T)$ are the depth values of the triangle vertices and $z_{far}$ is a large number representing infinity. The vertex attributes of a triangle $T$ are $a_i(T)$. The closest triangle index is computed as $T_c^*(\boldsymbol{p}) = \arg\min_T z_c(\boldsymbol{p}, T)$, and it determines which triangle is rendered for the attribute $a_c(\boldsymbol{p})$ in the crisp image. The soft image is computed as

$$z_s(\boldsymbol{p}, T) = \mathbf{1}\{0 < d(\boldsymbol{p}, T) < B\} \sum_{i=1}^{3} b_i(\boldsymbol{p}^*(T), T) z_i(T) + \tag{9}$$
$$(1 - \mathbf{1}\{0 < d(\boldsymbol{p}, T) < B\}) z_{far} + \lambda_{slope} d(\boldsymbol{p}, T)$$

$$a_s(\boldsymbol{p}) = \frac{\sum_T \mathbf{1}\{z_s(\boldsymbol{p}, T) < z_c(\boldsymbol{p}, T_c^*(\boldsymbol{p}))\} \sum_{i=1}^{3} b_i(\boldsymbol{p}^*(T), T) a_i(T)}{\sum_T \mathbf{1}\{z_s(\boldsymbol{p}, T) < z_c(\boldsymbol{p}, T_c^*(\boldsymbol{p}))\}} \tag{10}$$

$$\alpha_s(\boldsymbol{p}) = \max_T \mathbf{1}\{z_s(\boldsymbol{p}, T) < z_c(\boldsymbol{p}, T_c^*(\boldsymbol{p}))\}(1 - d(\boldsymbol{p}, T))/B, \tag{11}$$

where $B$ is the width of the extension around the triangle and $\lambda_{slope} > 0$. The final image $a(\boldsymbol{p})$ is computed as

$$a(\boldsymbol{p}) = \alpha_s(\boldsymbol{p}) a_s(\boldsymbol{p}) + (1 - \alpha_s(\boldsymbol{p})) \alpha_c(\boldsymbol{p}) a_c(\boldsymbol{p}) + (1 - \alpha_c(\boldsymbol{p})) \text{bcg}(\boldsymbol{p}), \tag{12}$$

where bcg is the background crop. UV mapping is supported as well: first the UV coordinates are rendered for the crisp and soft image. Then the colors are sampled from the texture map for the soft and the crisp image separately. Finally, the soft and the crisp images are blended. Figure 3 shows an illustration of the blended rendering as well as its effect on the training.

## 6 3D REPRESENTATION

The 3D representation $\mathbf{m} = [\mathbf{s}, \mathbf{t}, \mathbf{b}]$ consists of three parts, where $\mathbf{s}$ denotes the 3D shape of the object, $\mathbf{t}$ are the texture and $\mathbf{b}$ are the background colour values.

The shape $\mathbf{s}$ is a 3 dimensional array with the size of $3 \times N \times N$. We call this the shape image, where each pixel corresponds to a vertex in our 3D mesh and the pixel value determines the 3D coordinates

of the vertex. The triangles are defined as the subset of the regular triangular mesh on the $N \times N$ grid; we only keep the triangles of the middle circular region. The texture (image) is an array of the size $3 \times N_t \times N_t$. The renderer uses the UV mapping technique, so the size of the texture image can have higher resolution than the shape image. In practice we choose $N_t = 2N$, so the triangles are roughly 1 or 2 pixels wide and the texture can match the image resolution when rendering a $N \times N$ image. The background is a color image of size $3 \times 2N \times 2N$.

The renderer uses a perspective camera model, where the camera is pointing at the origin and placed along the $Z$ axis such that the field of view is set so the unit ball fits tightly in the rendered image. The viewpoint change is interpreted as rotating the object in 3D space, while the camera stays still. Finally a random $N \times N$ section of the renderer is cropped and put behind the object.

Notice that the 3D representation (3 dimensional arrays) are a perfect match for convolutional neural network generators. We designed it this way, so we can use StyleGAN as generator (see Karras et al. (2019)).

## 7 NETWORK ARCHITECTURE

We use the StyleGAN generator of Karras et al. (2019) with almost vanilla settings to generate the shape image, texture image and the background. StyleGAN consist of two networks: an 8 layer fully connected mapping network that produces a style vector from the latent inputs, and a synthesis network that consist of convolutional and upsampling layers that produces the output image. The input of the synthesis network is constant and the activations at each layer are perturbed by the style vector using adaptive instance normalization. For each layer activation also noise is added. It is also possible to mix styles from different latent vectors, by perturbing the convolutional layer activations with different styles at each layer. In our work we used the default setting for style mixing and for most parameters. We modified the number of output channels and the resolution and learning rates. The training was done in a progressively growing fashion, starting at the an image resolution of $N = 16$. The final resolution was set to $N = 128$.

One StyleGAN instance ($G_o$) generates the shape image and texture and another ($G_b$) generates the background. The inputs to both generators are 512 dimensional latent vectors $\mathbf{z}_o$ and $\mathbf{z}_b$. We sampled them independently, assuming the background and the object are independent. We set both $G_o$ and $G_b$ to produce images at $2N \times 2N$ resolution where $N$ is the rendered image size. The output of the object generator is then sliced into the shape and texture image, then the shape image is downsampled by a factor of 2. We multiplied the shape image by $0.002$, which effectively sets a relative learning rate, then added $\mathbf{s}_0$ to the output as an initial shape. For faces we set $\mathbf{s}_0$ to a sphere with a radius $r = 0.5$ and centered at the origin. For buses and cars $\mathbf{s}_0$ was a flat sheet.

We noticed that during the training the generation of shapes would not easily recover from bad local minima, which resulted in high frequency artifacts and the hollow-mask ambiguity. Thus we use a shape image pyramid to tackle this problem. The generator is set to produce $K = 4$ shape images, then these images are blurred with varying amounts and summed:

$$\mathbf{s}_{pyr} = \sum_{k=0}^{K-1} \frac{\mathrm{blur}(\mathbf{s}_i, \sigma = 2^k)}{2^k}, \tag{13}$$

where $\mathrm{blur}(\cdot)$ is the Gaussian blur and $\sigma$ is interpreted in pixels on the shape image.

We also noticed that the 3D models of the object tended to grow large and tried to model the background. This is the result of an acceptable ambiguity in the parametrization. In terms of the GAN objective it does not matter if the background is modelled by $\mathbf{b}$ or $\mathbf{s}$ and $\mathbf{t}$. As we are interested in results where the object fits in the image, we added a size constraint on the object. The output coordinates are computed as

$$\mathbf{s}_{size} = \mathbf{s} \frac{\tanh(|\mathbf{s}|)}{|\mathbf{s}|} s_{max}, \tag{14}$$

where we set $s_{max} = 1.3$ and the $L^2$ norm and $\tanh$ functions are interpreted pixel-wise on the shape image $\mathbf{s}$. The effect of both the shape image pyramid and the size constraint can be seen in Figure 4.

The discriminator architecture was also taken from StyleGAN and we trained it with default settings.

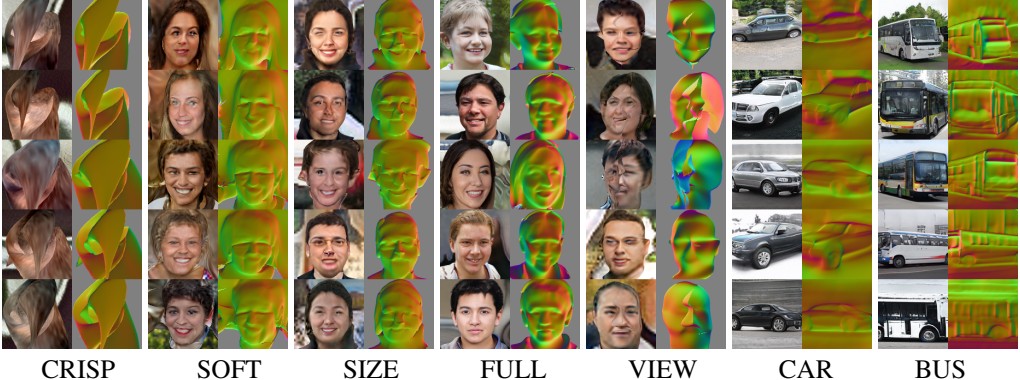

| CRISP | SOFT | SIZE | FULL | VIEW | CAR | BUS |

Figure 4: Samples from our methods. CRISP used the crisp renderer, while SOFT uses our proposed renderer. SIZE adds a size constraint in order to prevent the mesh modelling the background. FULL is our final model that adds the shape pyramid parametrization. CAR and BUS uses the same settings as FULL except it is initialized with a flat sheet instead of a sphere and we did not add size contraint.

Table 2: Ablations of our method. We show the different options and FID numbers on FFHQ and LSUN datasets.

| Method | Options | | | | | FID | | |
|---|---|---|---|---|---|---|---|---|
| | Renderer | Size C. | Pyramid | Init | View | Face | Car | Bus |
| CRISP | | | | sphere | ±45 | 377.6 | | |
| SOFT | ✓ | | | sphere | ±45 | 100.2 | | |
| SIZE | ✓ | ✓ | | sphere | ±45 | 94.6 | | |
| FULL | ✓ | ✓ | ✓ | sphere | ±45 | 63.3 | | |
| VIEW | ✓ | ✓ | ✓ | sphere | ±120 | 141.8 | | |
| CAR and BUS | ✓ | | ✓ | sheet | ±45 | | 59.6 | 65.3 |

# 8 EXPERIMENTS

We trained our model on the FFHQ faces (Karras et al., 2019) and on LSUN cars and buses Yu et al. (2015). FFHQ contains 70k high resolution ($1024 \times 1024$) colour images. We resized the images to $128 \times 128$ and trained our generative model on all images in the dataset. For the viewpoint we found that randomly rotating in the range of $\pm45$ degrees along the vertical and $\pm15$ degrees along the horizontal axis yielded the best results. This is not surprising, as most faces in the dataset are close to frontal. For our full model we used pyramid shapes of $4$ levels and size constraint of $1.3$ and we trained our model for 5M iterations. The results are shown on Figure 1 and more samples can be found in the Appendix. We also trained our model on $100k$ images from each of the LSUN categories. We used the same settings as for FFHQ except for the initialization of the 3D shape and without a size constraint.

In Figure 4 we show ablations of the choices of the renderer and network architecture settings. We can see that our soft renderer has a large impact on the training. The crisp renderer cannot learn the shape. Furthermore we can see that the size constraint prevents the mesh to model the background, and the shape pyramid reduces the artifacts and folds on the face. We can also see that it is important that we set the viewpoint distribution accurately. With the a large viewpoint range of $\pm120$ degrees our method struggles to learn. It produces self-intersecting meshes and puts multiple faces on the object. We can see that our method can also learn the 3D of other categories, such as cars and buses. Table 2 explains the options used in detail and shows quantitative results.

Figure 5 shows results on interpolated latent vectors. We can see the viewpoint and the identity is disentangled and the transition is smooth.

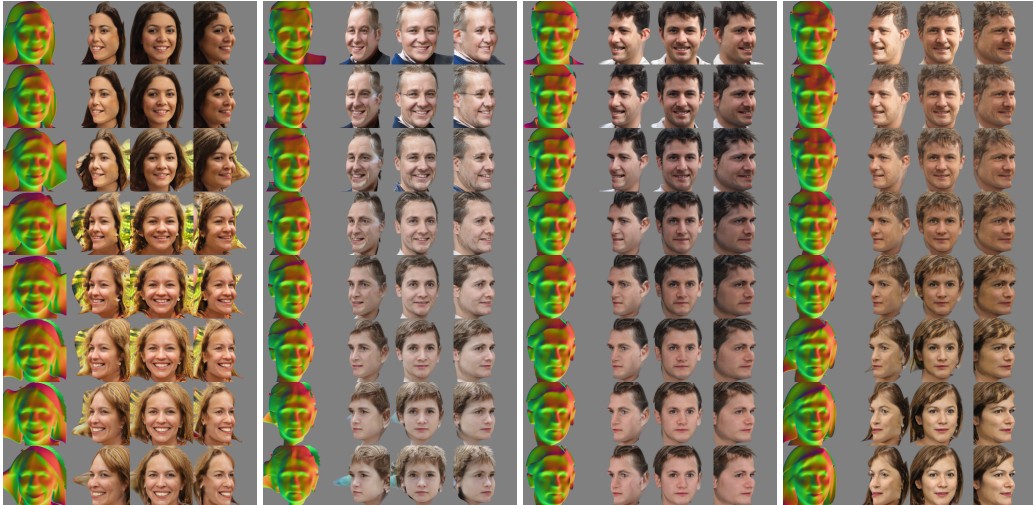

Figure 5: Interpolated 3D faces. From top to bottom the 3D models are generated by linearly interpolating the latent vector fed to the generator. We show 3 viewpoint in the range of $\pm 45$ degrees

## 9    DISCUSSION

Here, we would like to acknowledge the limitations of our work:

- Currently, we use a triangle mesh with fixed topology. In general, this is not sufficient for modeling challenging objects and scenes of arbitrary topology.

- The background is currently not modeled as part of the triangle mesh, which limits our method for datasets where the object is found at the center of the image. Note that this limitation is the result of the specific parametrization and the architecture of the generator and not of the expressive power of our method in general.

- The imaging model is currently a Lambertian surface illuminated with ambient light. However, specularity and directional light can be added to the renderer model. This is a relatively simple extension, but the representation of lights as random variables during training needs extensive experimental evaluation.

One might claim that our work does use supervision, as the faces FFHQ dataset were carefully aligned to the center of the images. We argue that our method still does not use any explicit supervision signal other than the images themselves. Moreover, as discussed in the limitation section, the centering of the object will be irrelevant when the background and the object share the same 3D mesh and texture. In contrast, methods that use annotation cannot be extended to deal with more challenging datasets, where that annotation is not available.

Another point is the motivation to generate faces in an unsupervised manner, since there already exist several data sets with lots of annotation. First, we choose FFHQ because it is a very clean dataset and our intention is to demonstrate that unsupervised 3D shape learning is possible. Second, we believe that unsupervised learning is the right thing to do even if annotation is available. Unsupervised methods can be extended to other datasets where that annotation is not available.

In conclusion, we provide a solution to the challenging and fundamental problem of building a generative model of 3D shapes in an unsupervised way. We explore the ambiguities present in this task and provide remedies for them. Our analysis highlights the limitations of our approach and sets the direction for future work.

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

# A    SAMPLES

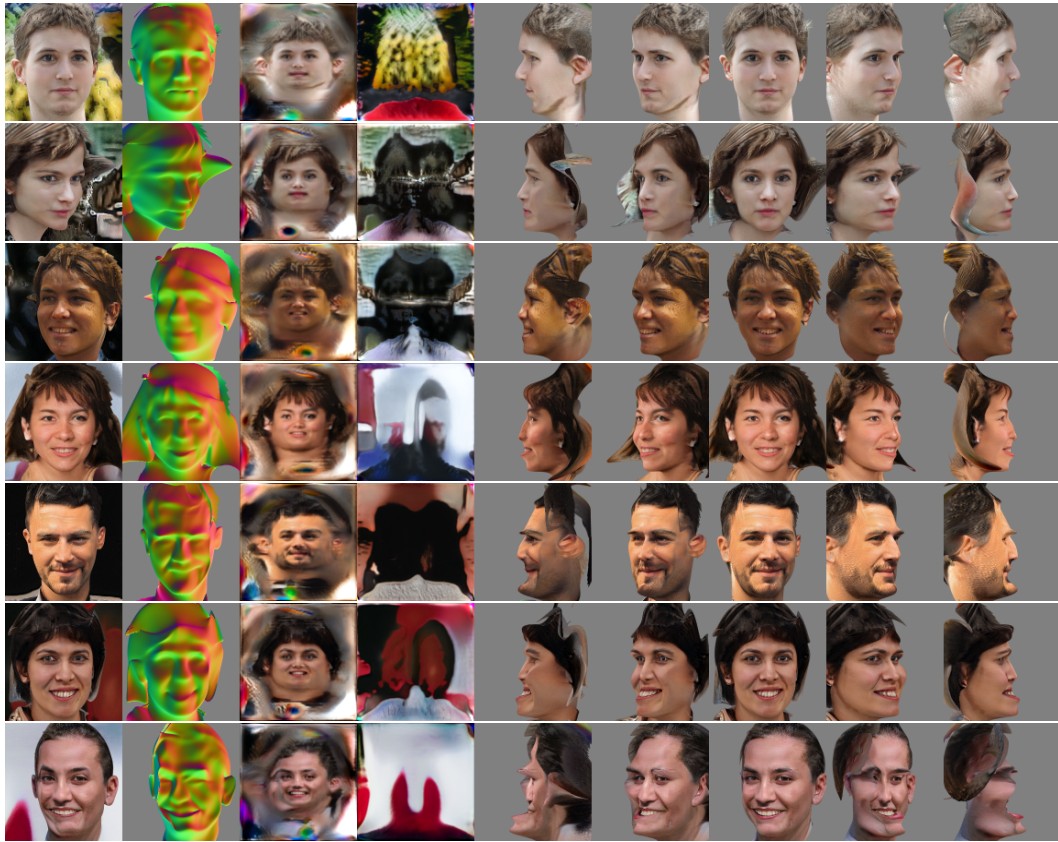

Figure 6: Samples from our generator trained on the FFHQ dataset at $128 \times 128$ resolution. The first column shows random rendered samples. The other columns show the 3D normal map, texture, background and textured 3D shapes for 5 canonical viewpoints in the range of $\pm 90$ degrees. The images were picked to illustrate the range of quality achieved by our method. We can see that most of the samples have anatomically correct shapes. In some cases there are exaggerated features like a large chin, that is only apparent from the profile view. Faces from those viewpoints are not present in the dataset, which might explain the shortcomings of our method. There are failure cases as well (the bottom row), that look realistic from the frontal view, but do not look like a face from the side.

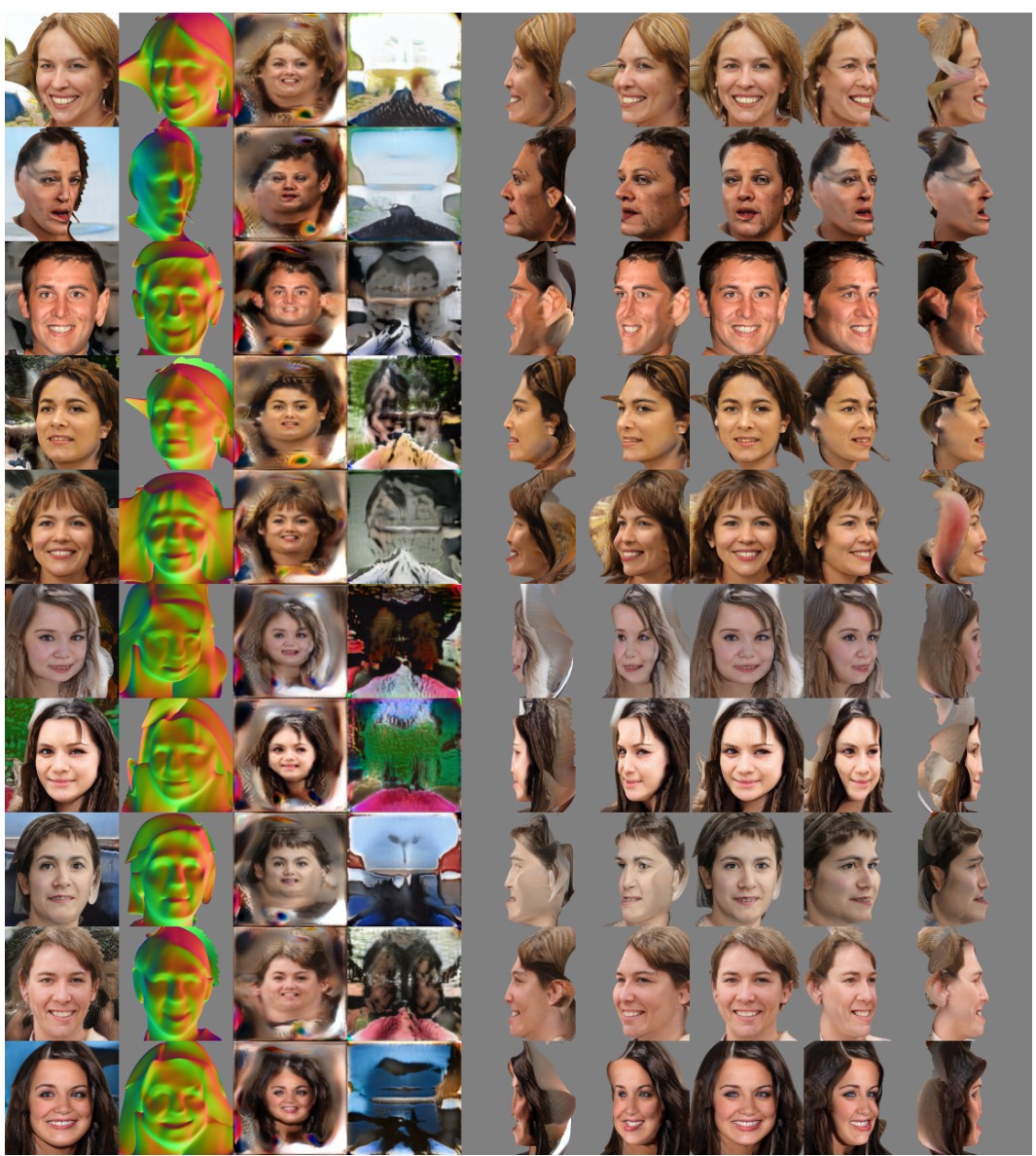

Figure 7: Random samples from our generator. The format is the same as in Figure 1.

