# OpenReview forum: "Unsupervised Generative 3D Shape Learning from Natural Images"
_ICLR.cc/2020/Conference — Reject_

### Official Review · AnonReviewer1 · 2019-10-11
**Official Blind Review #1**

**Rating:** 3

**Review:**

The paper tries to solve the problem of recovering the 3D structure from 2D images. To this end, it describes a GAN-type model for generating realistic images, where the generator disentangles shape, texture, and background. Most notably, the shape is represented in three dimensions as a mesh made of triangles. The final image is rendered by a fixed differentiable renderer from a randomly-sampled viewpoint. This allows the model to learn to generate realistic 3D shapes, even though it is trained only using 2D images.

The authors introduce a novel renderer based on the Lambertian image model, which is differentiable not only with respect to the texture but also with respect to position on mesh vertices, which allows better shape learning compared to prior art. Authors also identify some learning ambiguities: objects can be represented by the background layers, and sometimes object surface contains high-frequency errors. These are addressed by generating a bigger background and randomly selecting its part as the actual background, and by averaging shapes generated at different scales to smoothen the surface of generated objects, respectively. Authors do mention pitfalls of the model in the conclusion: fixed-topology mesh, the background is not modelled as a mesh, the model works only with images containing a single centred object, the image model is Lambertian.

I think that the approach is extremely interesting, addresses an important problem, and shows promising results. However, I vote to REJECT this paper, because the evaluation is insufficient, and the paper lacks clarity.

The approach is evaluated only on a single dataset and is not compared to any baselines. While results from ablations of the model are provided, they are only qualitative, consisting of a single example per ablation, and are hard to read and interpret---in particular, the provided description of the ablations and corresponding results is unclear. There are no quantitative results in the paper, and it is difficult for me to judge how good the method is given only qualitative examples from a single dataset.

As for clarity, I think that the distinction between supervised, unsupervised and weakly supervised learning in section 2 in unnecessary, does not add value to the paper, and can confuse the reader. Section 4 contains some unnecessary assumptions and incorrect claims. For example, the renderer R doesn't need to be able to generate perfect images for the approach to work; I think Theorem 1 is also incorrect since it does not take e.g. mode collapse into account, which prevents the learned distribution from being the same as the data distribution. Section 5 is very unclear, with practically no explanation for equations (6-11), which makes them very difficult to decipher.

The related works section is quite thorough, but the authors missed two extremely relevant papers: [1] and [2], which do a very similar thing and contain some of the ideas used in this paper.

I think the paper would be very valuable if the differentiable renderer was clearly explained and more evaluation and comparisons with baselines were provided.

[1] Rezende et. al., "Unsupervised Learning of 3D Structure from Images", NIPS 2016.
[2] Nugyen-Phuoc et. al., "HoloGAN: Unsupervised learning of 3D representations from natural images", ICCV 2019.


=== UPDATE ===
I appreciate adding more examples for ablations, FID scores and the LSUN dataset experiments. However, I still think that the exposition could be significantly improved, as eg. the description of the differentiable renderer is difficult to follow -- the equations should be better explained. Also, Figure 3. is difficult to understand; and the caption saying that "one rectangle overlaps another" is not helpful.

I think this is really cool work, but due to the lack of clarity, I think it shouldn't be accepted at this conference. Having said that, I am increasing my score to "weak reject" because of the improvements.


**Experience Assessment:**

I have read many papers in this area.

**Review Assessment: Checking Correctness Of Derivations And Theory:**

I carefully checked the derivations and theory.

**Review Assessment: Checking Correctness Of Experiments:**

I carefully checked the experiments.

**Review Assessment: Thoroughness In Paper Reading:**

I read the paper thoroughly.

---

> ### Author Response · Authors · 2019-11-14
> **answers**
>
> "only on a single dataset"
> We added results on LSUN categories.
>
> "not compared to any baselines"
> We are the fist one to solve this problem, thus a baseline is not available.
>
> "only qualitative, consisting of a single example per ablation ... corresponding results is unclear"
> We added detailed ablations and quantitative evaluation and explanations.
>
> "the distinction between supervised, unsupervised and weakly supervised learning in section 2 in unnecessary"
> We find it necessary as the term "unsupervised" is used in many different ways in the literature.
> By clearly defining the terms we highlight that:
> - the problem we solve is more difficult than the problems solved by prior work
> - based on our stricter definition of supervision signal, we provide the first "fully" unsupervised solution
>
> "unnecessary assumptions ... the renderer R doesn't need to be able to generate perfect images"
> The theory investigates if the solution of the optimization task is the desired one.
> In our work this analysis requires an ideal renderer R.
> Once the theory has been proved, one may also achieve a suboptimal solution
> by using approximations (eg, an imperfect renderer).
>
> "incorrect claims ... Theorem 1 is also incorrect since it does not take e.g. mode collapse into account"
> Mode collapse is taken into account at page 5 under the Assumption 4 of the original submission.
> Assumption 4 states that the GAN training is perfect, \ie, there is no mode collapse, where the training objective
> is defined in terms of expectations and not in terms of the training data.
>
> "differentiable renderer ... comparisons with baselines"
>
> We added comparisons with the baseline crisp renderer, but further evaluations are beyond the scope of this paper.

---

### Official Review · AnonReviewer3 · 2019-10-21
**Official Blind Review #3**

**Rating:** 8

**Review:**

SUMMARY: A new modification to the rendering mechanism in a differentiable renderer  to generate 3D images, trained in an unsupervised manner on 2D images of faces

CLAIMS:
- train a generative model in an unsupervised way on 2D face images,
- by generating 3D representation of [shape, texture, background] and feeding that to a differentiable renderer
- modify the rendering mechanism to be differentiable wrt vertices of the triangular mesh (in addition to the texture)
- curb the problems of training by using shape image pyramid, object size constraint


LIT REVIEW:
Well done, sufficient summarization of past work. But not "first":
it would be pertinent to mention the ICCV 2019 paper "HoloGAN: UNSUPERVISED LEARNING OF 3D REPRESENTATIONS FROM NATURAL IMAGES" (https://arxiv.org/abs/1904.01326), which tackles the exact same problem. It does not use a triangular mesh representation, or a differentiable renderer, instead it uses a 3D feature representation and a neural renderer. However, the problems tackled are very close to avoid mentioning this paper.

Hence, it might not be good to claim
- in the abstract: "the first method to learn a generative model of 3D shapes from natural images in a fully unsupervised way",
- in introduction: "For the first time, to the best of our knowledge, we provide a procedure to build a generative model that learns explicit 3D representations in an unsupervised way from natural images",
- and similar claims in other places.

Also, Pix2Scene (https://openreview.net/forum?id=BJeem3C9F7) also has similar ideas, although they tackled primitive shapes and not faces.

DECISION: This paper has very promising results.
Although it is limited to faces, which the community knows is something GANs are good at modeling because of the inherent structure, it is nevertheless a relevant piece of work in modelling 3D scenes in a graphics way and then training using adversarial learning. I am particularly impressed by the renderings of the depth and texture, and would be interested to explore that area further.

However, it is more pertinent to check how the model performs objects more complicated than faces. A very simple experiment is to try this on ImageNet images, which are also centered and aligned. This would help investigate the possibility of extending this method to more complicated objects than faces.

I would suggest to maybe put more focus on the fact that you have used the traditional graphics pipeline and integrated that into adversarial learning, as opposed to dealing with just weights and biases. That is indeed significant (in my opinion).

Knowing that most GAN training time is spent in overcoming a lot of failures, it would be great if the authors can summarize the failure cases and elaborate on the experiments performed to overcome those failures. This was briefly touched upon in Section 7, but it would be great if they could elaborate more on them possibly in the appendix.

It would be great if the authors can share their code, there was no mention of any possibility of this.

ADDITIONAL FEEDBACK:
Page 5:
...an added constrain*T* on m in the optimization...
...fooled by an inverted dept*H* image...
page 6:
...rendered <remove>attribute and</remove> depth, attribute and alpha map...

**Experience Assessment:**

I have published one or two papers in this area.

**Review Assessment: Checking Correctness Of Derivations And Theory:**

I assessed the sensibility of the derivations and theory.

**Review Assessment: Checking Correctness Of Experiments:**

I assessed the sensibility of the experiments.

**Review Assessment: Thoroughness In Paper Reading:**

I read the paper thoroughly.

---

> ### Author Response · Authors · 2019-11-14
> **answers**
>
> - HoloGAN's representation is latent, and not explicit, thus they do not output any 3D surfaces, unlike our model.
> - We added results on LSUN categories.
> - We added more ablation studies.
> - We will share our code upon publication

---

### Official Review · AnonReviewer2 · 2019-10-22
**Official Blind Review #2**

**Rating:** 3

**Review:**

I thank the authors for the rebuttal and the additional experiments. The additions do partially address my concerns, although not entirely. For instance, the experiments on non-face classes are very preliminary and it is unclear if they work at all (no other views shown). I hope the authors are right that the method will work on other classes after some tuning, but this is not demonstrated in the paper. Overall, I am quite in a borderline mode. I think the paper looks promising and after further improving the experimental evaluation it can become a great publication. But for now the experiments, especially the new ones, look somewhat incomplete and rushed, more suitable for a workshop paper. Therefore, I still lean towards rejection.

---

The paper proposes an approach to learning the 3D structure of images without explicit supervision. The proposed model is a Generative Adversarial Network (GAN) with an appropriate task-specific structure: instead of generating an image directly with a deep network, three intermediate outputs are generated first and then processed by a differentiable renderer. The three outputs are the 3D geometry of the object (represented by a mesh in this work), the texture of the object, and the background image. The final output of the model is produced by rendering the geometry with the texture and overlaying on top of the background. The whole system can be trained end-to-end with a standard GAN objective. The method is applied to the FFHQ dataset of face images, where it produces qualitatively reasonable results.

I am in the borderline mode about this paper. On one hand, I believe the task of unsupervised learning 3D from 2D is interesting and important, and the paper makes an interesting contribution in this direction. On the other hand, the experimental evaluation is quite limited: the results are purely qualitative, on a single dataset, and do not contain much analysis of the method. It would be great if the authors could add more experiments to the paper during the discussion phase.

More detailed comments:
Pros:
1) The paper is presented well, is easy to read. I like the detailed table with comparison to related works, and a good discussion of the limitations of the method and the tricks involved in making it work. I also like section 4 clearly discussing the assumptions of the work, although I think it could be shortened quite a bit.
2) The proposed method is reasonable and seems to work in practice, judging from the qualitative results.

cons:
1) The experiments are limited.
1a) There are no quantitative results. I understand it is non-trivial to evaluate the method on 3D reconstruction, although one could either train a network inverting the generator, or, perhaps simpler, apply a pre-trained image-to-3D network to the generated images. But at least some image quality measures (FID, IS) could be reported.
1b) The method is only trained on one dataset of faces. It would be great to apply the method to several other datasets as well, for instance, cars, bedrooms, animal faces, ShapeNet objects. This would showcase the generality of the approach. Otherwise, I am worried the method is fragile and only applies to very clean and simple data. Also, if the method is only applied to faces, it makes sense to mention faces in the title.
1c) It would be very helpful to have more analysis of the different variants of the method, ideally with quantitative results (again, at least some image quality results). Figure 3 goes in this direction, but it is very small and does not give a clear understanding of the relative performance of diferent variants.

2) A missing very relevant citation of HoloGAN by Nguyen-Phuoc et al.  [1]. It is not yet published, but has been on arXiv for some time. I am a bit unsure about the ICLR policy in this case (this page https://iclr.cc/Conferences/2019/Reviewer_Guidelines suggests that arXiv paper may be formally considered prior work, in which case it should be discussed in full detail), but at least a brief mention would definitely be good.

[1] HoloGAN: Unsupervised learning of 3D representations from natural images. Thu Nguyen-Phuoc, Chuan Li, Lucas Theis, Christian Richardt, Yong-Liang Yang. arXiv 2019.

**Experience Assessment:**

I have published one or two papers in this area.

**Review Assessment: Checking Correctness Of Derivations And Theory:**

I assessed the sensibility of the derivations and theory.

**Review Assessment: Checking Correctness Of Experiments:**

I carefully checked the experiments.

**Review Assessment: Thoroughness In Paper Reading:**

I read the paper at least twice and used my best judgement in assessing the paper.

---

> ### Author Response · Authors · 2019-11-14
> **answers**
>
> - We provide FID numbers in the revised paper.
> - We provide results on LSUN categories. The results are not as good as on the faces, but:
> 	- we did not have enough time to tune the algorithm to this new data,
> 	  but we believe that eventually it would work well also here as we observe similar
>           challenges to the faces dataset at the early stages of the algorithm development
> 	- LSUN includes outliers, cropped objects, occlusions, high variation in shape,
> 	  position and scale
> - We added detailed ablations.
> - We added HoloGAN to the related work (also see my answer to all reviewers and the updated paper)

---

### Public Comment · ~Bernhard_Egger1 · 2019-09-30
**initialization and literature**

Just some comments, not a proper review:
This paper looks very interesting and novel!
I did not read it in full detail yet - my main question would be in the direction of how the learning is initialized?
And, can you reasonably interpolate between instances? Does the generator learn meaningful correspondences?

I think some of the following works might be worth adding to the table?
- Thomas J Cashman and Andrew W Fitzgibbon. 2012. What shape are dolphins? building 3d morphable models from 2d images. IEEE Transactions on Pattern Analysis and Machine Intelligence 35, 1 (2012), 232–244.

- Ayush Tewari, Florian Bernard, Pablo Garrido, Gaurav Bharaj, Mohamed Elgharib,
Hans-Peter Seidel, Patrick Pérez, Michael Zollhoefer, and Christian Theobalt. 2019.
FML: Face Model Learning from Videos. In Proc. IEEE Conference on Computer Vision
and Pattern Recognition (CVPR).

- Luan Tran, Feng Liu, and Xiaoming Liu. 2019. Towards High-fidelity Nonlinear 3D
Face Morphable Model. In Proc. IEEE Conference on Computer Vision and Pattern
Recognition (CVPR).

---

> ### Author Response · Authors · 2019-10-01
> **answers**
>
> Dear Bernhard Egger,
>
> Thank you for your interest in our work. You can find our answers below
>
> - Initialization: You can find the answer in my other comment.
>
> - Correspondences: One can establish semantic correspondences by using the vertex IDs, i.e the same vertex should represent for example the corner of the eye in multiple generated samples. However these correspondences are not guaranteed by the theory because of the ambiguity in the parametrization. When interpolating between different samples, the location of semantic features might move smoothly from one vertex to another. The random samples show that in practice the vertices are more or less aligned with the semantic keypoints, as the the same facial features are generated on the same places on the texture map across instances.
> One can test how well our method learns correspondences quantitatively by inverting the generator (and renderer) and backproject the ground truth keypoint locations to the mesh. Inverting the generator is a work in progress, and we aim to show results on correspondences in future work.
>
> - Related work: Thank you for pointing us to these relevant papers.

---

> > ### Public Comment · ~Bernhard_Egger1 · 2019-10-01
> > **Thank you**
> >
> > Thanks for the clarification - great work!

---

### Author Response · Authors · 2019-10-01
**some details and fixed typos**

Dear Readers and Reviewers,

We would like to correct some typos and add some details we left out from the paper.

- Section 5: The fixed renderer equations are:

\begin{align}
z_c(\vp, T) = &~\textstyle \1\{d(\vp, T)=0\} \sum_{i=1}^3 b_i(\vp,T) z_i(T) + (1-\1\{d(\vp, T)=0\}) z_{far} \\
a_c(\vp) = &~ \textstyle\1\{ z_c(\vp, T_c^*(\vp)) < z_{far})\} \sum_{i=1}^3 b_i(\vp,T^*_c(\vp)) a_i(T^*_c(\vp)) \\
\alpha_c(\vp) = &~  \textstyle\1\{ z_c(\vp, T_c^*(\vp)) < z_{far})\}
\end{align}
\begin{align}
z_s(\vp, T) =~& \textstyle \1\{0<d(\vp, T)<B\}  \sum_{i=1}^3 b_i(\vp^*(T),T) z_i(T) + \\
& \quad\quad\quad(1-\1\{ 0 < d(\vp, T)<B\}) z_{far} + \lambda_{slope} d(\vp, T) \nonumber\\
a_s(\vp) =~&  \frac{\sum_{T} \1 \{ z_s(\vp, T) < z_c(\vp, T^*_c(\vp)) \} \sum_{i=1}^3 b_i(\vp^*(T),T) a_i(T)  }
{ \sum_{T} \1 \{ z_s(\vp, T) < z_c(\vp, T^*_c(\vp)) \} } \\
\alpha_s(\vp) =~&  \textstyle\max_{T} \1 \{ z_s(\vp, T) < z_c(\vp, T^*_c(\vp)) \} (1-d(\vp,T) )/B,
\end{align}

- Section 7: the shape image is downsampled, not the texture image
- Section 7: the discriminator architecture is the vanilla StyleGAN and trained with default settings.
- Section 7: Initialization: We multiplied the shape image (the raw output of the generator) by 0.002, which effectively sets a relative learning rate, then added s_0 to the output as an initial shape. We set s_0 to a sphere with a radius r = 0.5 and centered at the origin.

---

### Public Comment · ~Tushar_Jain1 · 2019-10-15
**Quantitative Evaluation and comparison with prior work?**

Hi there,

Is there any Quantitative Evaluation (some evaluation metric) and comparison with prior work?

Thanks.

---

> ### Author Response · Authors · 2019-10-16
> **answers**
>
>
> Dear Tushar Jain,
>
> Thank you for your question. Indeed, it is unusual to leave out quantitative comparisons from a machine learning paper. We will explain our reasons below.
>
> Quantitative evaluations:
> - direct 3D evaluation: Our model can be evaluated quantitatively in 3D reconstruction tasks and semantic key-point matching once it is inverted. In our case this means we should find a mapping from images to the latent representation after which the generator reconstructs the 3D shape.
> Inverting our generative model is out of the scope of this paper as it is not a straightforward task and it needs extensive experimental evaluation. We aim to do that in the future using the insights of the latest research papers on inverting GANs. Then we can compare our method with many of the inverted 3DMM models and other supervised techniques.
>
> - indirect 2D rendering evaluation: We did not include 2D metrics in the evaluation because our main focus was learning the 3D shape of objects in the training set. We can add inception score and FID in the final version and compare our work with 2D generative models.
>
> Comparisons with prior work:
> In Table 1 we compare our method with prior SOTA in terms of supervision signals and capabilities. We did not compare our method in terms of performance metrics for the reasons discussed above. Also note that our work is the first unsupervised 3D method for natural images, so it would be unfair to expect it to perform at the level of SOTA supervised methods.

---

### Author Response · Authors · 2019-11-14
**detailed answers to reviewers**


Dear Reviewers,

Thank you for your feedback. Before we address the reviewers concerns, we summarize our contributions:
1) We address the problem of 3D shape learning from natural images in a fully unsupervised way.
This has never been solved before in our general settings (below we clarify in detail the reviewers concerns on this claim).
2) Experiments show our method produces high quality results, which demonstrates the feasibility of
this novel task (in the revised paper we provide additional experiments that reviewers requested).
3) We provide theoretical analysis to formalize the conditions under which 3D can be learned without supervision in our settings.

Therefore, we believe that our paper would be of great interest to the ICLR community.

Here, I address the main complaints:

1) NOVELTY:
There are many papers on "unsupervised" learning, generative 3D models and disentangling. But none of them works under our general settings.
We added the mentioned relevant papers to the table:
[1] Rezende et. al., "Unsupervised Learning of 3D Structure from Images", NIPS 2016.
	- they learn 3D of synthetic images (we do it for natural images)
[2] Rajeswar et. al. "Pix2Scene: Learning Implicit 3D Representations from Images"
	- they learn 3D of synthetic images (we do it for natural images)
[3] Nguyen-Phuoc, et. al., "HoloGAN: Unsupervised learning of 3D representations from natural images", ICCV 2019.
	- they do disentangling of viewpoint and object, which allows rendering the object from different viewpoints
	- their model does not learn an explicit 3D shape representation, but a latent one
	- their model does not provide a depth map or normals, and cannot be readily used in traditional graphics pipeline
	- their model does not guarantee that the rendered views are consistent, i.e. there exists a 3D object that has those views
	- the efficacy of their method has not been proven theoretically

2) EXPERIMENTS:
We provide the requested (new) experiments:
- FID numbers of the several generators
- More detailed ablations (samples in the supplementary)
- a trained model on 2 LSUN categories

3) THEORY:
While empirical evidence is necessary to validate an approach, theory is fundamental to point out
shortcomings of a method and therefore its further development.
In practical terms, our analysis was paramount in showing us that the method could potentially work
and in showing us how to handle the viewpoint selection and the background layer.

---

### Decision · Program_Chairs · 2019-12-19

**Decision:**

Reject

**Comment:**

The paper proposes a GAN approach for unsupervised learning of 3d object shapes from natural images. The key idea is a two-stage generative process where the 3d shape is first generated and then rendered to pixel-level images. While the experimental results are promising, the experimental results are mostly focused on faces (that are well aligned and share roughly similar 3d structures across the dataset). Results on other categories are preliminary and limited, so it's unclear how well the proposed method will work for more general domains. In addition, comparison to the existing baselines (e.g., HoloGAN; Pix2Scene; Rezende et al., 2016) is missing. Overall, further improvements are needed to be acceptable for ICLR.


Extra note: Missing citation to a relevant work
Wang and Gupta, Generative Image Modeling using Style and Structure Adversarial Networks
https://arxiv.org/abs/1603.05631